# Research Progress of Porcine Reproductive and Respiratory Syndrome Virus NSP2 Protein

**DOI:** 10.3390/v15122310

**Published:** 2023-11-24

**Authors:** Benjin Liu, Lingzhi Luo, Ziqi Shi, Houbin Ju, Lingxue Yu, Guoxin Li, Jin Cui

**Affiliations:** 1College of Veterinary Medicine, Northeast Agricultural University, Harbin 150038, China; b220602021@neau.edu.cn (B.L.); b230602013@neau.edu.cn (L.L.); b230601006@neau.edu.cn (Z.S.); 2Shanghai Animal Disease Prevention and Control Center, Shanghai 201103, China; chengzinono@126.com; 3Shanghai Veterinary Research Institute, Chinese Academy of Agricultural Sciences, Shanghai 200241, China; guoxinli@shvri.ac.cn

**Keywords:** PRRSV, NSP2, −2/−1 programmed ribosomal frameshifting, Interferon, autophagy, apoptosis

## Abstract

Porcine reproductive and respiratory syndrome virus (PRRSV) is globally prevalent and seriously harms the economic efficiency of pig farming. Because of its immunosuppression and high incidence of mutant recombination, PRRSV poses a great challenge for disease prevention and control. Nonstructural protein 2 (NSP2) is the most variable functional protein in the PRRSV genome and can generate NSP2N and NSP2TF variants due to programmed ribosomal frameshifts. These variants are broad and complex in function and play key roles in numerous aspects of viral protein maturation, viral particle assembly, regulation of immunity, autophagy, apoptosis, cell cycle and cell morphology. In this paper, we review the structural composition, programmed ribosomal frameshift and biological properties of NSP2 to facilitate basic research on PRRSV and to provide theoretical support for disease prevention and control and therapeutic drug development.

## 1. Introduction

### 1.1. Epidemiology of PRRSV

Porcine reproductive and respiratory syndrome virus (PRRSV) is the causative pathogen of porcine reproductive and respiratory syndrome (PRRS) and is characterized by reproductive disorders in sows and respiratory disorders in pigs of all ages. The disease has been prevalent around the world for more than 30 years and has seriously harmed the global pig breeding industry. In 1992, the World Organization for Animal Health (OIE) classified PRRSV as a Category B infectious disease [1]. Currently, PRRSV is divided into PRRSV-1 and PRRSV-2 [2,3], of which PRRSV-2 is the dominant strain in China. The first PRRSV strain, CH-1a, was isolated in China in 1996 [4]. In 2006, HP-PRRSV (a highly pathogenic PRRS virus) broke out in China, and its virus genome is characterized by 30 amino acid deletions in the NSP2 hypervariable region [5]. Subsequently, China discovered the main prevalent NADC30 strain in China in 2015, and the potentially prevalent NADC34 strain in China in 2018 [6,7,8,9] (Table 1). Beginning in 2020, a severe PRRSV outbreak in northeastern Spain was characterized by a high rate of abortion and increased mortality among weaners. Virus isolation and sequencing analysis showed that they were caused by a mosaic PRRSV strain, derived from the PR40 strain reported by Canelli, after drifting and undergoing several recombination events with other PRRSV-1.1 isolates. The strains quickly changed from an acute course to a slow form in evolution, and the virulence was weakened [10,11]. In addition, research has shown that all respiratory and intestinal mixed infections lead to increased farm losses, which is exacerbated by secondary infections with Streptococcus suis and intestinal bacteria [12]. PRRSV continues to evolve new variants with increasingly complex differentiation and virulence phenotypes [13], posing enormous challenges to disease prevention and control.

### 1.2. PRRSV Viral Genome Structure and Function

PRRSV is a member of the Arteriviridae family and is an enveloped single-stranded positive-sense RNA virus [21]. Its genome is approximately 15 kb and contains at least 10 known open reading frames, ORF1a, ORF1b, ORF2a, ORF2b, ORF3-7 and ORF5a, as well as flanking 5′ and 3′-untranslated regions (Figure 1) [22]. Except for ORF1, the other open reading frames are translated into viral structural proteins and participate in virus assembly; ORF1 is approximately 12 kb long, accounting for approximately 80% of the viral genome and encodes two larger subreading frames, namely ORFla and ORFlb. The polypeptides produced by translation are hydrolyzed and processed by autoencoded proteases to produce at least 16 nonstructural proteins (NSP1-12), which together with structural proteins, are involved in regulating host cells (Table 2) [21,23].

PRRSV has the characteristics of easy mutation, strain diversity, immunosuppression, difficulty in producing neutralizing antibodies and ADCC effects. In recent years, China has approved many attenuated vaccine strains for use, making the virus prone to mutation, resulting in frequent recombination between field virus strains and vaccine strains and the emergence of a series of new PRRSV mutant strains, such as NADC30-like or NADC34-like strains [6,7,9]. The emergence of new strains and the characteristics of the virus itself have made the prevention and control of PRRSV very challenging.

## 2. NSP2 Structural Composition and Ribosome Frameshifting

### 2.1. Structural Composition and Genetic Variation of NSP2

NSP2 is the largest replicase cleavage product of PRRSV and is a large multidomain protein of 1066–1197 aa. The structural composition consists of six domains, including the N-terminal hypervariable region HV-Ⅰ, cysteine protease PL2 region, Peptidase_C33 structure, 500–700 amino acid hypervariable region HV-Ⅱ and transmembrane domain TM. NSP2 has a hydrophilic cytoplasmic domain tail composed of approximately 200 amino acids at the C-terminus. Because NSP2 has a hypervariable region, the amino acid similarity of NSP2 in strains of the same subtype is 66–70%, but the similarity in strains of different subtypes is only approximately 30% [20]. Therefore, NSP2 has the largest variation in the genome [69], and NSP2 also has the ability to recombine between strains: in vivo recombination experiments found that the NSP2, ORF3 and ORF5 genes will recombine and exchange when different strains infect SPF pigs at the same time [70].

The PLP2 domain of NSP2 has cis and trans cleavage activities. At least six NSP2 isoforms of different lengths can be generated through enzymatic cleavage: NSP2a, NSP2b, NSP2c, NSP2d, NSP2e and NSP2f. Their expression levels are relatively stable. Among these isoforms, NSP2d, NSP2e and NSP2f cleavage sites are all located in the hypervariable region. There are no predictable conserved enzyme cleavage sites, and they have no obvious impact on PRRSV replication [27]. Studies have confirmed that the PLP2 domain can inactivate cis and trans cleavage activities through C55A and H124C [71]. In addition, the plp2 domain also has deubiquitination and deacylation functions, which can be lost through site mutations such as D85R or D91R [72].

The hypervariable region of NSP2 is flexible and contains multiple B-cell epitopes. Jun Han et al. [73] used VR2332 as the backbone through a reverse genetic system to construct a series of mutant strains and indicated that NSP2 can be randomly deleted between 324 and 813 (aa), and it was found that a maximum of 400 aa (NSP2 324–726) can be deleted and 13–35 aa at the N end can be deleted but other positions cannot be deleted. When the mutant strain has fewer deletions, it has growth kinetics and RNA expression profiles similar to those of the parent virus, but the lytic activity of the NSP2 324–726 mutant on MARC-145 cells is reduced, and no visible plaques appear. This finding indicates that the caspase domain PL2 of NSP2, its downstream Peptidase_C33 structure, the putative transmembrane domain and the C-terminal tail region are necessary for replication. The deletable positions were mainly concentrated in the B-cell epitope region in the NSP2 hypervariable region. This region may be used as a bait epitope to deviate the host immune response from key viral proteins and to interfere with the host’s immune response. However, this region cannot be completely deleted. It may be crucial to maintain the spatial structure of the entire protein. In addition, Jun Han et al. also found that by inserting the gene encoding GFP into a NSP2 deletion mutant, the constructed recombinant virus was damaged and unstable and could gradually gain parental growth ability by losing most of the GFP gene [73]. The ability of the virus to quickly be lost is amazing.

Reviewing previous studies, the characteristics of mutable deletions in the hypervariable region of NSP2 are consistent with the HP-PRRSV outbreak in China in 2006, which had a 30-amino-acid deletion in the B-cell epitope region of NSP2, but the deletion in this region has been shown not to be associated with changes in virulence [74]. This is also consistent with the structural understanding of NSP2. Little is known about other structures of NSP2, especially the function of its C-terminus, and more important biological functions of NSP2 need to be elucidated.

### 2.2. NSP2 and Ribosome Frameshifting

Programmed ribosomal frameshift (PRF) is a mechanism by which host cells or viruses produce multiple proteins through overlapping reading frames. PRF was first proposed in sarcoid viruses, which generate gag–pol fusion proteins from transiently overlapping gag and pol ORFS by sliding the host cell ribosome backward one position, −1 PRF, during translation of the viral RNA genome [75,76,77]. Arter viruses such as PRRSV have evolved a −2/−1 PRF mechanism. PRRSV has two PRF sites, one of which is located at the end of ORF1a and generates ORF1ab through −1 PRF; the other PRF site is clever, the −1 and −2 PRF events occurs at the same site in the NSP2 coding region of ORF1a, resulting in two NSP2 variants. Respectively, it occurs in −1 PRF and generates a truncated structure compared to that of the full-length NSP2 called NSP2N, while the structure generated in −2 PRF contains an alternative C-terminal domain called NSP2TF [78,79].

At the core of most −1 PRFs are ribosomes interacting with a stimulatory mRNA structure (stem-loop or RNA pseudoknot), which promotes translational frameshifting of a stretch of polybases called a slippery sequence. PRRSV −2/−1 PRF is a highly efficient −2/−1 PRF event that functions without identifiable stimulating RNA secondary structure [78,79]. Specifically, pcbp interacts with PRRSV mRNA through KH1 and KH3, with each domain linked to one of two C-patches (CCCA and CUCC) in the C-rich region, which is located in the sliding sequence (GG GUU UUU) approximately 10 bp downstream. The KH2 domain binds to NSP1b instead of viral RNA, and the complex disrupts the normal sliding of ribosomes. The NSP1b-PCBP complex causes ribosome stalling, possibly as a result of direct interaction with the unwinding center of the small subunit, in a manner similar to that described for structured RNA stimuli [80]. An increasing number of similar mechanisms have been discovered, including Stm1 of *Saccharomyces cerevisiae*, which is a fragile, thereby inhibiting translation [81,82,83].

The −1/−2 PRF of NSP2 has important biological significance for PRRSV. The recombinant virus that silences NSP2TF severely affects virus replication and produces small plaques but is not lethal [78,84]. Traditionally, it has been shown that approximately 20% of ribosomes frameshift the translation of NSP2 into the −2 reading frame, thereby producing a transframe fusion protein (NSP2TF), and 7% of ribosomes enter the −1 reading frame, where they immediately encounter a stop codon, thus synthesizing a truncated version of NSP2 called NSP2N [79]. Georgia M. Cook found, through RNA seq methods, that −2 PRF efficiency, which may be promoted by PRF stimulating the accumulation of the viral protein NSP1β during infection, increased over time [85]. Before 2011, almost all PRRSV would immediately encounter the stop codon of NSP2 −1 PRF and stop transcription. However, the percentage of stop codon mutant strains increased rapidly after 2011. From 2011 to 2021, as many as 50% of the sequences lacked stop codons at this position. These dominant mutant strains came from farms where PRRSV outbreaks had occurred and were on the rise, such as pig farms [84]. Reverse genetics approaches were used to construct a mutant strain using SD9521 as the viral framework and VR2332 sliding cassette. The results of the Slippage to Construct box experiment showed that mutant strains carrying UGG or CGG at the −1 PRF site of the NSP2 gene had higher virus titers and higher −2 PRF efficiency. Surprisingly, two −1 PRF stop codon mutation patterns were identified to be predominantly prevalent in the field. These mutation patterns show higher growth kinetics than other mutant strains, indicating that the most dominant −1 PRF stop codon mutation pattern may enhance the growth adaptability of the virus [84], and we speculate that the mechanism may be different for base pair effects on the different binding abilities and gliding efficiencies of the NSP1b-PCBP complex and the small subunit. This strategy of PRRSV using PRF to produce polyproteins is important as it allows for viruses with smaller genomes to produce more proteins to regulate viral replication.

## 3. The Function of NSP2

### 3.1. The Relationship between NSP2 and Virus Assembly

NSP2 plays an important role in viral assembly [86,87]. PRRSV ORF1 translationally produces long-chain polypeptides, which need to be sheared to form polysaccharides to function and complete viral assembly. The plp2 functional domain of NSP2, which has both cis and trans cleavage activity, cleaves both its own and nonentropic polypeptides to form multiple NSP2 subunits (NSP2a, NSP2b, NSP2c, NSP2d, NSP2e and np2f) or is responsible for cleavage between NSP2 and NSP3. In addition, NSP2 cooperates with NSP4 to complete cleavage between NSP4 and NSP5. In addition, the NSP2 essential variant, NSP2TF, interacts with two major viral envelope proteins (GP5 glycoprotein and membrane (M) protein) to drive key processes in arterivirus assembly and outgrowth, while NSP2TF targets the cytosolic pathway to reduce the proteasome-driven turnover of the GP5/M protein, thereby facilitating the formation of the GP5-M dimer, which is essential for arterivirus assembly [86]. NSP2N and NSP2 may not share this function due to differences in subcellular localization or three-dimensional structure.

PRRSV is a positive-stranded RNA virus, and a key step in its replication is the assembly of the replication and transcription complex (RTC) [88], while the virus modifies the intracellular environment to generate bilayer vesicles that provide a barrier for itself. In addition, viruses also utilize aggregates (aggresomes), which provide a rich material base for viral replication and assembly, while being able to evade recognition by the immune system. In the arteritis virus EAV, NSP2 and NSP3 are involved in the formation of the RTC, a multiprotein complex that facilitates viral replication to modify cell membranes. NSP2 interacts with NSP3 to form replicative bilayer vesicles, which promote the formation of aggresomes [89,90]. PRRSV NSP2 and NSP3 have also been shown to interact [91]. It has been shown that PRRSV NSP2 recruits the Bcl2-associated antiapoptotic gene 6 (BAG6) protein to help localize in the endoplasmic reticulum, thereby promoting the formation of bilayer membrane vesicle structures. In addition, NSP2 binds to the cytosolic protein 14-3-3, an important protein involved in aggregate formation, through the highly variable region or the C-terminal tail region to promote aggresome formation, enhancing viral replication [87]. In addition, NSP2 has also been shown to be present among purified virus particles, but its specific function has not been elucidated. Therefore, NSP2 may be involved in PRRSV replication as a structural protein in addition to being a nonstructural protein.

### 3.2. The Relationship between NSP2 and Immunology

#### 3.2.1. The Relationship between NSP2 and Innate Immunity

Natural immunity is important as the first line of host defense against viral infection. The intracellular RLR receptor RIG-I/MDA5 is an important molecule for the recognition of RNA viruses (Figure 2). RIG-I senses and binds to viral 5′ppp-RNA through the C-terminal CTD structural domain, undergoes a conformational change and releases the CARD structural domain, which in turn activates RIG-I via catalytic polyubiquitination of the K63 linkage of the CARD and CTD by ubiquitin ligases such as TRIM25, TRIM4 and Riplet (RNF135). RIG-I is activated by catalyzing the polyubiquitination of the K63 linkage of CARD and CTD. The zinc finger protein ZCCHC3 can act as a coreceptor for RLR to facilitate the binding of RIG-I and MDA5 to viral RNA and can promote polyubiquitination and activation of the K63 linkage of the CARD structural domains of RIG-I and MDA5 by TRIM25 [92]. The PRRSV NSP2 protein inhibits the ubiquitination activation of RIG-I (DDX58) by means of TRIM25 through plp2 deubiquitination activity, and ZCCHC3 knockdown also promotes PRRSV replication [92].

The structural domains of CARDs form tetramers after RIG-I activation and recruit MAVS proteins located in the outer mitochondrial membrane through CARD-CARD interactions. MAVS dimerizes and forms a complex with DDX3X and TRAF3, which enhances the activation of intrinsic immunity to anti-RNA viruses and in turn activates TBK1 and the IRF3 protein through the phosphorylation of TBK1. Phosphorylated IRF3 forms a dimer, enters the nucleus, binds to the promoter region of the IFN gene and induces the production of interferon; in addition, activated MAVS can also degrade IKBα through ubiquitination, and P65 enters the nucleus to induce the expression of antiviral proteins. The N-terminal cysteine protease domain of NSP2 attenuates the interaction of MAVS with the deconvolution enzyme DDX3X by interacting with the N-terminal DExDc and C-terminal HELICc domains of the enzyme. The interaction with MAVS is weakened by NSP2, which affects the level of MAVS-mediated IRF3 phosphorylation and prevents activation of the IFN promoter, thus antagonizing the antiviral effect of DDX3X. In addition, NSP2 inhibits the NF-KB pathway and inflammatory factor expression by deubiquitinating K48-linked IKBα, preventing its ubiquitination and degradation, and thereby retaining P65 in the cytoplasm [93]. Whether NSP2N and NSP2TF bind to DDX3X to inhibit IRF3 phosphorylation or deubiquitinate IKBα to suppress the expression of inflammatory factors has not been reported.

After interferon production, P65 is secreted outside the cell and binds to the surface receptor JAK in neighboring cells. The cytoplasmic region of the JAK kinase is phosphorylated and activated. STAT is transferred to a position near the JAK kinase and undergoes tyrosine phosphorylation. The tyrosine phosphorylated STAT starts to detach from the IFN receptor and forms a homodimer or a heterodimer, and the STAT1-2 heterodimer binds to IRF9 to form the activated transcriptional complex ISGF3. The STAT homodimer itself is a transcriptional activator, and the dimer transfers to the nucleus and induces the production of the antiviral protein ISG. ISG15 is an important antiviral protein that can exert antiviral effects through multimerization of the substrate, and it can also promote STAT1/STAT2 activation and ISGF3 formation through the interaction of free ISG15 with STAT2, which in turn promotes ISGF3-induced ISRE activation to exert free-state ISG15 antiviral effects [94]. NSP2 inhibits the expression and coupling of the antiviral protein ISG15 through the PLP2-DUB structural domain [93], resulting in the premature detachment of ISG15 from key signaling molecules. This mechanism acts as an emergency immune evasion mode for PRRSV, silencing the inflammatory response and IFN response [95], which in turn inhibits the amplification of IFN to adjacent cells. Recently, it was found that ISG15 acts as a substrate for TRIM21 glycosylation to enhance p62 ubiquitination to prevent autophagosome formation [96], and it is interesting to see whether NSP2 removes the inhibitory effect of autophagy via this pathway, although autophagy has been reported to be favorable in some reports [97]. There are no relevant reports on the role of NSP2N and NSP2TF in ISG15.

#### 3.2.2. The Relationship between NSP2 and Acquired Immunity

NSP2, which contains B-cell epitopes and predicted T-cell epitopes, is an immunogenic protein of PRRSV [98], and high levels of antibodies comparable to or even higher than those against the N protein have been measured in porcine blood from one week to several months after viral infection [99,100,101,102]. Six of the ten linear B-cell epitopes contained in type 1 PRRSV were found to be present in NSP2 using phage display [98], and eighteen B-cell epitopes were found in NSP2 using peptide scanning, two of which are located at the position of deletions in the highly variable region of HP-PRRSV NSP2 [100]. Some researchers used GOLDKEY (IBMS, AMMS, Ürümqi, China) and DNASTAR software (Inc., Madison, WI, USA) to predict the dominant epitopes of NSP2 in BJ-4 and used peptide scanning to find the presence of six B-cellular epitopes at amino acids 73-567, of which SP4, SP5, SP6 and SP8 are immunodominant epitopes [103]. The same T-cell epitopes were found to exist for NSP2 by analysis and prediction [69]. Since NSP2 can be detected in purified virus particles, NSP2 may be present in virus particles as a structural protein, which may be the key reason why NSP2 antibodies are present at high levels in infected pig sera for a long period of time [104]. NSP2N and NSP2TF share the N-terminal region of NSP2 and therefore share all of the above characteristics as well.

The major histocompatibility complex (MHC) is a group of tightly interlinked genes encoding major histocompatibility antigens, i.e., leukocyte antigens. Porcine SLA-DR, the porcine leukocyte antigen, plays an important role in immunity, and the OTU domain of PRRSV NSP2 acts as a deubiquitinating enzyme, blocking ubiquitination and proteasome-mediated degradation of the α- and β-chains of SLA-DR and leading to the accumulation of SLA-DR proteins in PRRSV-infected cells. Naturally, PRRSV infections often result in SLA-DR-mediated NSP protein presentation and corresponding antibodies. NSP protein presentation and corresponding antibody production, thus, affect organism-specific immunity [93]. This ability of NSP2 may be responsible for the massive production of non-neutralized antibodies against PRRSV.

Since the emergence of PRRS in 1987, even though scientists have made continuous efforts to understand the pathogenesis and vaccinology of PRRSV, vaccines have not been successfully developed to prevent PRRSV widely and effectively, MLV is still the vaccine mainly used for prevention and control [105]. NSP2 is closely related to virulence and immunity and is one of the focuses of PRRSV vaccine research [106,107,108]. Earlier, the PRRSV TJM vaccine strain was found to have a sequential 120-amino-acid deletion (NSP2 628–747 aa) after a 30-amino-acid deletion in viral culture, which was thought to be related to its decreased virulence [102]. Subsequently, residues 500–596 and 658–777 of HP-PRRSV NSP2 were found to be required for the up-regulation of COX-2 expression and PGE2 production. The mutant HP-PRRS virus with residue 500–596 or 658–777 deletions had impaired ability to up-regulate COX-2 and PGE2 production in vitro and in vivo. Importantly, pigs infected with the variant virus can alleviate fever, clinical symptoms and mortality, providing clues for the development of an attenuated HP-PRRSV vaccine [109]. In fact, as early as 2009, some authors tried to construct a mutant NSP2 attenuated vaccine and proved that NSP2 is an important target for the development of tagged vaccines and virus attenuation [110]. Subsequently, two recombinant viruses that knocked out the ribosomal frameshift variants of NSP2, NSP2TF and NSP2N were shown to promote IFN-α responses, improve NK cell cytotoxicity and enhance T cell immune responses in infected pigs [105]. In addition, peptides 562–627 (m1B) and 749–813 (m2B) of the NSP2 protein were recently found to have no significant effect on viral replication. Therefore, m1B and m2B have the potential to be molecular markers for PRRSV vaccines [111]. Recently, the papasin-like protease 2 (PLP2) domain of PRRSV NSP2 was shown to be a target for neutralizing antibodies, and genetic variation in NSP2 resulted in poor cross-neutralization. When animals were immunized with the NSP2 chimeric strain, they were resistant to challenges with the NSP2 parent strain, but not with the backbone virus. In addition, neither the immune strain nor the challenge strain could be detected in vaccine-immunized animals, highlighting the important role of HP-PRRSV NSP2 in promoting viral clearance [112]. In conclusion, with a better understanding of NSP2 function, partial loss or retention of NSP2 will present opportunities for MLV vaccines.

### 3.3. The Relationship between NSP2 and Autophagy

Autophagy is a physiological process of material degradation–recycling through the lysosomal degradation of damaged organelles or proteins in eukaryotic cells [113], and it is also involved in the infection process of many viruses [114,115,116]. PRRSV can promote its own replication by inducing autophagy (Figure 3) [117,118].

Autophagy as an important physiological process in eukaryotic cells is negatively regulated through the PI3K-AKT-mTOR and MAPK-ERK-mTOR inactivation of ULK1 [119,120,121]. Therefore, autophagy is negatively regulated, or positively regulated through AMPK-ULK1. The researchers found that PRRSV infection induced autophagy in the following ways: PRRSV infection activates the opening of endoplasmic reticulum IP3R channels, allows Ca^2+^ into the cytoplasm, causes Ca^2+^ imbalance in the ER, induces endoplasmic reticulum stress, increases endoplasmic reticulum stress marker protein GRP78, activates the MAPK and PI3K signaling pathways, and promotes autophagy formation. On the other hand, endoplasmic reticulum calcium ion levels were reduced by the endoplasmic reticulum Ca^2+^ receptor STIM1, which accumulates and translocates to the vicinity of the plasma membrane [117]. The accumulated STIM1 protein then activates Orai1, prompting the opening of the CRAC calcium channel pore, extracellular calcium inward flow to the endoplasmic reticulum, endoplasmic reticulum Ca^2+^ influx into the cytoplasm via IP3R again and cytoplasmic Ca^2+^ level increases [122]. Calcium is a ubiquitous intracellular messenger involved in a variety of signaling processes, such as gene transcription, differentiation, proliferation and kinase activation [13]. Ca^2+^ and Ca^2+^ receptor proteins play important roles in the regulation of autophagy, and excessive release of Ca^2+^ from the endoplasmic reticulum activates the calmodulin-dependent protein kinase ii (CaMKII) and adenosine monophosphate-activated protein kinase (AMPK) signaling cascades and increases the lipidation of LC3II, leading to the inhibition of mTOR signaling and the induction of autophagy [123,124]. In addition, the ER is an important organelle for Ca^2+^ storage and protein processing and is highly sensitive to viral infection [125,126]. When endoplasmic reticulum function is disrupted, misfolded and unfolded, proteins accumulate in the lumen of the endoplasmic reticulum, interfering with Ca^2+^ homeostasis and leading to endoplasmic reticulum stress, which in turn induces autophagy [126]. PRRSV infection induces endoplasmic reticulum stress and disrupts Ca^2+^ homeostasis by means of a mechanism that ultimately activates autophagy through the CaMKII-AMPK-mTOR pathway in order to promote autophagy for self-replication [117]. CaMKII (a multifunctional Ca^2+^-dependent kinase) is activated in response to elevated Ca^2+^ levels, which in turn phosphorylates the downstream effector AMPK and enhances the lipidation of LC3II to promote autophagy. This process can be inhibited by 4-PBA, which suppresses PRRSV replication by inhibiting endoplasmic reticulum activation, and by ML-9 HCL, which suppresses viral replication by inhibiting SOCE channels [127]. The authors found that NSP2 interacts with STIM1 and GRP78 to induce autophagy by overexpressing Marc145 cells through a single gene [117]. The effect of NSP2 on the STIM1 and GRP78 proteins has not yet been demonstrated. Meanwhile, PRRSV-induced endoplasmic reticulum stress opens the SOCE channel to allow extracellular Ca^2+^ to enter the intracellular space. The SOCE channel is the main calcium channel on the plasma membrane of the cell, which mediates Ca^2+^ entry from the extracellular medium. Orai1 on the plasma membrane and STIM1 on the endoplasmic reticulum are the proteins responsible for SOCE channel activation [117].

In addition, it has been shown that NSP2 can activate the endoplasmic reticulum stress pathway by decreasing BAG6 expression through the interaction with BAG6, which affects the localization of NSP2 in the cytoplasm [128]. NSP2 binds to BAG6 and protects itself from ubiquitinated degradation through plp2 activity, and mutations in plp2(CH/AA) activity on NSP2 are ubiquitinated and degraded by BAG6. PLP2 activity also inhibits the ubiquitinated degradation of error proteins by BAG6, which in turn activates endoplasmic reticulum stress [128]. Interestingly, endoplasmic reticulum Ca^2+^ levels showed a decreasing trend at 2–4 h of viral infection, whereas NSP2 was gradually expressed after 4 h [129]. Therefore, the initial activation of IP3R channels may not be induced by NSP2, such as lymphocytes being activated by antigens and consequently Ca^2+^ translocation to the cytoplasm [130]. The reason why NSP2 expression alone can also induce endoplasmic reticulum stress may be closely related to the fact that NSP2 is rich in B-cell and T-cell epitopes. Recently, it has been found that cells can use autophagy to inhibit PRRSV NSP1α protein expression to suppress viral replication [97], which is contrary to the conclusion that autophagy promotes PRRSV replication, and it is clear that this game of virus–host interaction is complex and seemingly contradictory.

### 3.4. The Relationship between NSP2 and Apoptosis

Apoptosis is the programmed cell death of multicellular organisms that occurs in an autonomous and orderly manner, under certain physiological and pathological conditions, and is controlled by their own genes to maintain the stability of their own internal environment [131]. Viral infection and apoptosis have been hot topics of research, and PRRSV infection can cause apoptosis in a variety of organs, tissues and cells in pigs, such as the lungs, lymphatic system, testes, placenta, etc. The infected cells are mainly macrophages, lymphocytes, germ cells and dendritic cells [132,133,134,135,136]. Recently, it has been reported in the literature that NSP2 induces apoptosis, which is second only to NSP4 in the intensity of apoptosis induced, but the mechanism is not clear [137]. It has been shown that NSP2 can inhibit the ubiquitin proteasome pathway degradation of AIF1, which in turn causes no caspase-dependent apoptosis through the endoplasmic reticulum stress pathway, and its binding ability can be abolished through mutation of the plp2 active site, in addition to the caspase-dependent apoptosis induced by NSP2 [128]. It has been shown that PRRSV-1 cannot cause caspase-independent apoptosis, which may be related to the large difference in NSP2 between the two [138].

CD2AP and SH3KBP1 belong to a group of “scaffolding molecules” that regulate the actin cytoskeleton and can affect endocytosis, cell death, cell adhesion and division through anchoring and remodeling of actin [139]. It was found that NSP2 binds to CD2AP and SH3KBP1 [87]. The biological significance of NSP2 binding to SH3KBP1 has not yet been clearly demonstrated. The NSP2 highly variable region proxylate motif (PXXXPR motif) interacts with the SH3 domain region of CD2AP, resulting in the aggregation of CD2AP around the nucleus, altered subcellular localization and induced CatL-dependent degradation of CD2AP [87]. On the one hand, the degradation of CD2AP TGF-b1 secretion is also promoted, which can inactivate the anti-regulatory signal TGF-b1-CD2AP-PI3K-ATK and enhance the pro-apoptotic signal TGF-b1-Smad3-P38 [140], but interestingly, PI3K-ATK is in an inhibited state only in the middle stage of PRRSV infection [141], which may be the result of the virus in the middle stage of infection in order to breaking the barrier effect of alveolar epithelial cells for invasion of the organism and cell-to-cell spreading. On the other hand, the degradation of CD2AP disrupts the cytoskeleton, leading to defective cell division and more cells arresting in the S phase [142]. In addition, it has been found that, in inflammation, macrophages deficient in CD2AP can lose their motility [143], which may attenuate the phagocytosis of PRRSV by macrophages. Thus, NSP2 plays a regulatory role in various aspects, such as apoptosis.

## 4. Conclusions

PRRSV, a virus with a long history and a serious threat to the global pig farming industry, has been the subject of extensive basic research. Important progress has been made in the treatment and prevention of PRRSV, also known as blue ear disease, but there is still no highly effective, cross-protective vaccine for clinical prevention. NSP2 plays an important role in the maturation of PRRSV proteins, assembly of viral particles, modulation of the host immune response, cellular autophagy, apoptosis, alteration of the cytoskeleton and regulation of the cell cycle (Figure 4). In addition, NSP2 uses −1/2 PRF to translate its own nucleic acid sequence into the NSP2N and NSP2TF variants, which have the same or similar functions in many aspects while retaining their own unique functions because all three of these proteins have identical amino-terminal domains, especially the PLP2 domain. It seems that −1/2 PRF can also be regarded as an opportunity to inhibit viral replication by blocking the −1/2 PRF site to achieve therapeutic effects. NSP2 of PRRSV is characterized by recombination, deletion and high mutation, and the mechanism of its −1/2 PRF site makes the study of the protein challenging. The complex and varied structural composition of NSP2 and its wide range of functional characteristics make the protein a mystery.

In this paper, we provide a more detailed depiction of the PRRSV NSP2 structure, function and genetic variation to provide a reference for basic research on PRRSV and to facilitate a deeper exploration of NSP2 function. We found that NSP2 functions are many and varied, but there are few studies related to the comparison of the functional variability of NSP2, NSP2N and NSP2TF. What are the roles played by each of them or synergistically, what are the functions unique to each of them and what are the shared functions, and what determines these functions? In addition, some studies have shown that NSP2 can also exist as a structural protein, but what is its specific functional significance? There are many unanswered questions that scientists need to continue to explore to solidify basic research on PRRSV and to fight for the prevention and control of blue ear disease and for the development of the global pig farming industry.

## Figures and Tables

**Figure 1 viruses-15-02310-f001:**
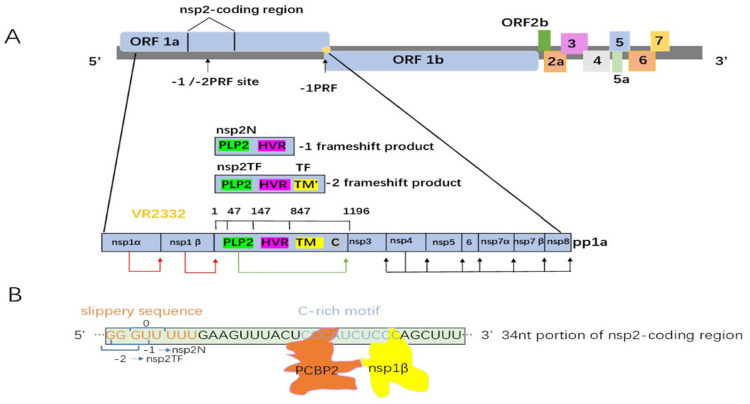
Map of the PRRSV genome. (**A**) The PRRSV genome is composed of ORF1a, ORF1b, ORF2a, ORF2b, ORF3–7 and ORF5a and produces at least 16 nonstructural and 8 structural proteins. NSP2 is cleaved from ORF1 translation products to form mature proteins and it is composed of HV-I, PLP2, HV-2, TM and C-terminal domains. (**B**) NSP2 undergoes a −1/2 ribosomal frameshift and generates NSP2N and NSP2TF variants in response to both pcbp2 and NSP1β.

**Figure 2 viruses-15-02310-f002:**
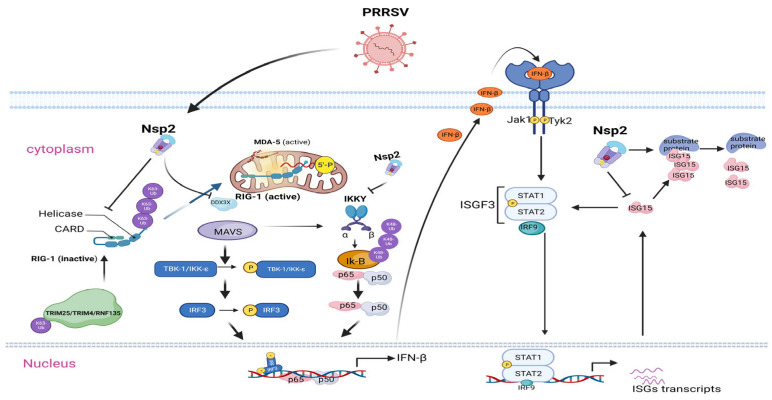
NSP2 regulates innate immunity. NSP2 antagonizes the innate immunity of host cells by deubiquitinating RIG-1 and IKBα or reducing the protein level of ISG15 or ISG15. NSP2 antagonizes host innate immunity by deubiquitinating RIG-1 and IKBα or separating ISG15 from its substrate and reducing ISG15 protein levels.

**Figure 3 viruses-15-02310-f003:**
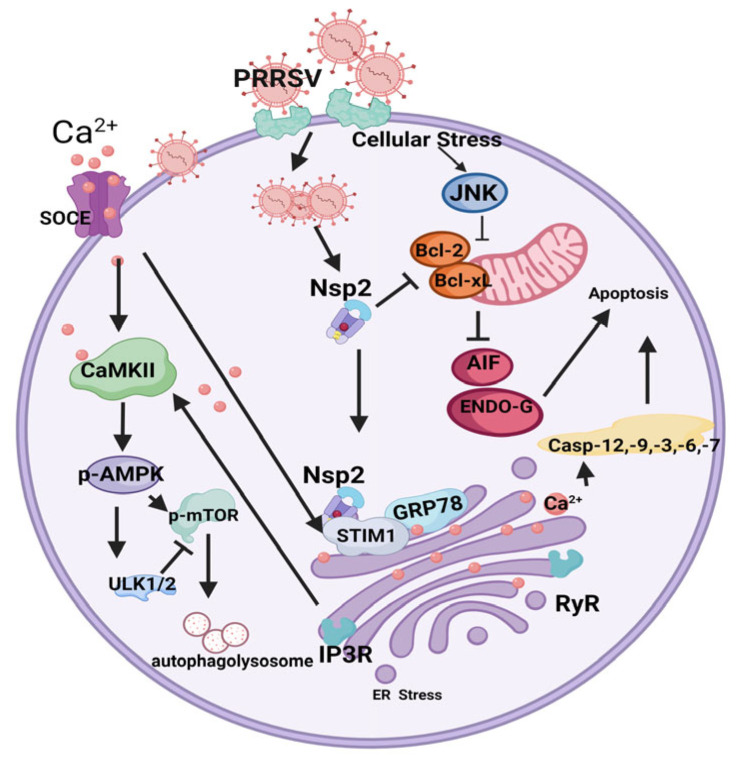
NSP2 regulates cell autophagy and apoptosis. NSP2 regulates endoplasmic reticulum stress and autophagy by binding to endoplasmic reticulum stress protein GRP78 and Ca^2+^ sensor protein STIM1. NSP2 binds to anti-apoptotic protein Bcl-2 and reduces its expression to induce apoptosis.

**Figure 4 viruses-15-02310-f004:**
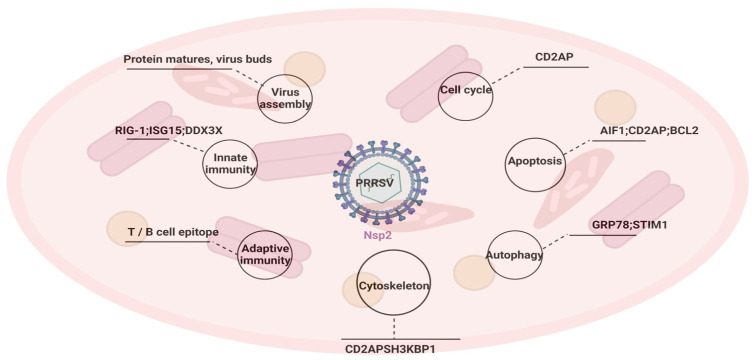
NSP2 of PRRSV regulates host cell and viral replication. NSP2 regulates cytoskeleton, cell cycle, autophagy, apoptosis, cellular innate immunity, viral protein maturation and viral assembly by reducing the expression of key proteins or modifying key proteins by means of deubiquitination.

**Table 1 viruses-15-02310-t001:** Important epidemic PRRSV strain information.

Years	Country	Strain	Strain Characteristics	References
1987	America	ATCC VR2332	PRRSV-2 prototype strain	[14]
1990	Europe	LV	PRRSV-1 prototype strain	[15]
1992	Japanese	GU992M	First reported in Asia	[16,17]
1990s	America	VR2385	A highly virulent PRRSV-2 strain identified in the mid-1990s	[18]
1995	China	CH-1a	First reported in China, the PRRSV pathogen was isolated by Guo et al. (1996) and Yang et al. (1997)	[4]
1998	America	Atypical PRRSV	Causes high fetal mortality and abortions in pig herds	[19]
2001	China	MN184	The highly virulent MN184 strain has very obvious nucleotide differences (>14.5%) from other genotype 2 strains	[20]
2006	China	JXA1	30-amino-acid deletion in NSP2 and highly pathogenic PRRSV (HP-PRRSV)	[5]
2015	China	NADC30-like	The current main circulating strains in China	[6,9]
2018	China	NADC34-like	Could become a potential pandemic strain in China	[7,8]

**Table 2 viruses-15-02310-t002:** Functions of each protein of PRRSV.

ORF	Protein	Function	References
ORF1a	NSP1 (α)	Loss of functional NSP1αpapain-like proteinase activity is associated with the inhibition of sg mRNA synthesis.	[24]
	NSP1 (β)	Loss of papain-like proteinase activity of NSP1β results in the NSP1β/NSP2 adapter not being processed in vitro.	[25,26]
	NSP2a/NSP2b/NSP2c/NSP2d/NSP2e//NSP2f/NSP2TF	NSP2 is a cofactor of nsp4 serine protease and participates in the assembly of polyproteins during virus replication.	[27]
	NSP3	NSP3 remodels the cell membrane.	[28,29]
	NSP4	NSP4 is a potential IFN antagonist that induces apoptosis.	[30,31,32,33,34]
	NSP5	NSP5 is a transmembrane protein involved in cell membrane modification.	[35]
	nsp6	It is involved in the virulence of the virus.	[36]
	NSP7α	It is one of the most conserved proteins in PRRSV, NSP7 and NSP7α/NSP7β and is critical for virus viability and recovery.	[37]
	NSP7β		
	NSP8	It is involved in the virulence of the virus.	[38]
ORF1ab	NSP9	It is a key protein for the high virulence of HP-PRRSV and the RdRp of PRRSV, which plays a key role in the virus replication process.	[39,40,41,42,43]
	NSP10	It is the helicase of PRRSV and is the key to unraveling dsRNA and RNA synthesis during PRRSV replication and ultimately completing virus replication.	[44]
	NSP11	It is the B cell linear epitope, has RNA endoribonuclease activity and deubiquitination activity.	[45,46,47]
	NSP12	It is necessary for the synthesis of viral subgenomic SG mRNA and promotes viral replication.	[48,49,50]
ORF2a	GP2a	Special glycosylated proteins capable of inducing specific neutralizing antibodies against PRRSV.	[51]
ORF2b	E	It is a pore-forming protein that mediates the formation of ion channels for viral particle uncoating.	[52,53,54]
ORF3	GP3	The main component of the viral envelope surface polymer promotes the early infection of highly pathogenic PRRSV in epithelial cells and can induce the production of neutralizing antibodies against PRRSV.	[55,56]
ORF4	GP4	The key adhesion protein of the PRRSV virus, the GP4 protein of PRRSV-1, has a neutralizing antibody-inducing epitope, which can induce the body to produce homologous strain-specific strong neutralizing antibodies.	[55,57,58]
ORF5	GP5	As the most variable transmembrane glycosylated protein among structural proteins, it has linear and conformational neutralizing antibody epitopes and is the most important protective anti-neutralizing antibody-inducing protein.	[59]
ORF6	M	It is the most conserved structural protein and contains a large number of antigenic epitopes. It has strong immunogenicity and can form heterodimers with GP5.	[60,61,62,63,64,65]
ORF7	N	Viral capsid protein promotes PRRSV RNA synthesis, contributes to PRRSV replication and increases PRRSV virulence.	[66,67,68]

## Data Availability

Not applicable.

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
