# Peer review of "Research Progress of Porcine Reproductive and Respiratory Syndrome Virus NSP2 Protein"

_viruses, 2023, doi:10.3390/v15122310_

Round 1
Reviewer 1 Report
Comments and Suggestions for Authors
I noticed that in some parts of the test there are too many spaces and in others there are none:
Abstract fisrt line after prevalent;
Introduction ( Table 1)
Paragraph 2.2. twelfth line pseudoknot) ,
lethal(75,81). tradionally
Paragraph 3.1. (83, 84).PRRSV
( NSP2a
(83) .
Paragraph 3.2.1. (88).PRRSV
TBK1.Phosphorylated
proteins.The
ISG.ISG15
(90).NSP2
Figure 2 IKBa or
Paragraph 3.3. [10].Ca2+
autophagy .PRRSV
demonstrated. , Meanwhile
4h.
Conclusions cycle. In addition
Some word in italic form: in vitro or in vivo (Table 2 and Paragraph 2.1.)
In paragraph 3.3. Ca2+ must be replaced with Ca2+
For the bibliography in the test i suggest to insert more references in the paragraphs 3.3. and 3.4.
In the paragraph 3.1. the name of autohr must be replaced with number of reference (Nan H, 2018)
In the figure 3 caption STIMI is must be replaced with STIM1 and in the figure 4 AIFI is must be replaced with AIF1, as in the text of paragraph 3.4.
Moreover, I'm not sure if the order of references is correct, please check the authors instructions.
Author Response
Dear editor and reviewers,
Thank you very much for the reviewing process of our manuscript entitled “Research Progress of Porcine Reproductive and Respiratory Syndrome Virus NSP2 Protein” (ID: viruses-2718368). We also highly appreciate the reviewer’s carefulness, conscientiousness, and beneficial suggestions. Those comments are all valuable and very helpful for revising and improving our paper, as well as guiding to our research. We have gone through the manuscript very carefully, and made some improvements as suggested in the revised manuscript. These changes will not influence the overall content and framework of the paper. We deeply appreciate your consideration of our manuscript, and we look forward to receiving comments. If we still have problems or faults, please tell us, we will try our best to revise. We appreciate for Editors/Reviewers’ warm work earnestly, and hope that the correction will meet with approval.
Specifically, We carefully corrected some white space, superscript, italics, and protein name writing problems, and re-screened and corrected the whole article. In addition, we inserted more references in paragraphs 3.3 and 3.4 to make our statements more evidence-based. Next, the name of the author of "(Nan H, 2018)" in paragraph 3.1 has been replaced with a reference number. Finally, we have checked the authors instructions, and there was also some adjustment in the order of the references due to the citation of the new references. The revised version includes newly introduced discussions and references, presented in the Word version of the manuscript in modified mode or marked in red. Please see the attachment.
Once again, thank you very much for your comments and suggestions.
Yours sincerely
Jin Cui
E-mail: jincui@neau.edu.cn

Reviewer 2 Report
Comments and Suggestions for Authors
Comments for the authors
Major comments
A. You should discuss the recent reports of a PRRSV strain of enhanced virulence in Europe (PRRSV L1C 1-2-4, Rosalia / Italy and Spain). Please send for example the following references
Martín-Valls, G.E., Cortey, M., Allepuz, A. et al. Introduction of a PRRSV-1 strain of increased virulence in a pig production structure in Spain: virus evolution and impact on production. Porc Health Manag 9, 1 (2023). https://doi.org/10.1186/s40813-022-00298-3
Claudia Romeo, Giovanni Parisio, Federico Scali, Matteo Tonni, Giovanni Santucci, Antonio M. Maisano, Ilaria Barbieri, M. Beatrice Boniotti, Tomasz Stadejek, G. Loris Alborali. Complex interplay between PRRSV-1 genetic diversity, coinfections and antimicrobial use influences performance parameters in post-weaning pigs. Veterinary Microbiology, Volume 284, 2023, 109830,
B. You should discuss the recent research on PRRSV vaccinology in relation to Nsp2 protein
Author Response
Dear editor and reviewers,
Thank you very much for the reviewing process of our manuscript entitled “Research Progress of Porcine Reproductive and Respiratory Syndrome Virus NSP2 Protein” (ID: viruses-2718368). We also highly appreciate the reviewer’s carefulness, conscientiousness, and beneficial suggestions. Those comments are all valuable and very helpful for revising and improving our paper, as well as guiding to our research. We have gone through the manuscript very carefully, and made some improvements as suggested in the revised manuscript. These changes will not influence the overall content and framework of the paper. We deeply appreciate your consideration of our manuscript, and we look forward to receiving comments. If we still have problems or faults, please tell us, we will try our best to revise. We appreciate for Editors/Reviewers’ warm work earnestly, and hope that the correction will meet with approval.
Responses to A:
Thank you very much for your comments. We have discussed the strain and cited the corresponding references in paragraph 1.1. Please see Line-13 of the paragraph 1.1 of manuscript.
Responses to B:
Thank you very much for raising an important knowledge point about nsp2, but we just missed it. In Section 3.2.2, we discuss the relationship between nsp2 and vaccines, and the corresponding references are attached, which enriches the content of our article. Thank you again for your valuable comments Please see the third paragraph of Section 3.2.2 of manuscript.
Once again, thank you very much for your comments and suggestions.
Yours sincerely Jin Cui E-mail: jincui@neau.edu.cn
